# Removal Performance and Mechanism of Benzo(*b*)Fluorathene Using MnO_2_ Nanoflower/Graphene Oxide Composites

**DOI:** 10.3390/ma14164402

**Published:** 2021-08-06

**Authors:** Qingqing Cao, Siqi Lu, Wenjun Yin, Yan Kang, Naihao Yang, Yudong Hou, Zizhang Guo

**Affiliations:** 1School of Architecture and Urban Planning, Shandong Jianzhu University, Jinan 250014, China; caoqingqing18@sdjzu.edu.cn; 2College of Environmental Science and Engineering, Tongji University, Shanghai 200092, China; 1851284@tongji.edu.cn; 3College of Environment and Safety Engineering, Qingdao University of Science and Technology, Qingdao 266042, China; kangyan@qust.edu.cn; 4Jinan Engineering Consulting Institute, Jinan 250014, China; qdhfx.238@163.com; 5Majian International Architectural Design Consulting Co., Ltd., Jinan 250014, China; hou.yu.dong@163.com; 6Shandong Key Laboratory of Water Pollution Control and Resource Reuse, School of Environmental Science and Engineering, Shandong University, Qingdao 266237, China

**Keywords:** MnO_2_ nanoflower, graphene oxide, PAHs, adsorption

## Abstract

High-ring polycyclic aromatic hydrocarbons (PAHs, Benzo[b]fluorathene (BbFA), etc.) are difficult to biodegrade in the water environment. To address this issue, an innovative method for the preparation of MnO_2_ nanoflower/graphene oxide composite (MnO_2_ NF/GO) was proposed for adsorption removal of BbFA. The physicochemical properties of MnO_2_ NF/GO were characterized by SEM, TEM, XRD, and N_2_ adsorption/desorption and XPS techniques. Results show that the MnO_2_ NF/GO had well-developed specific surface area and functional groups. Batch adsorption experiment results showed that adsorption capacity for BbFA was 74.07 mg/g. The pseudo-second-order kinetic model and Freundlich isotherm model are fitted well to the adsorption data. These show electron-donor-acceptor interaction; especially π-π interaction and π complexation played vital roles in BbFA removal onto MnO_2_ NF/GO. The study highlights the promising potential adsorbent for removal of PAHs.

## 1. Introduction

Polycyclic aromatic hydrocarbons (PAHs) refer to aromatic hydrocarbons containing two or more benzene rings which are formed by the incomplete combustion or pyrolysis of fossil fuels such as coal, oil and natural gas, wood, paper, and other hydrocarbons under reduced conditions [1]. The toxicity, genotoxicity, mutagenicity and carcinogenicity of PAHs cause a variety of harms to the human body, such as damage to the respiratory system, circulation system, nervous system, liver, and kidney damage [2,3]. PAHs have recently attracted much attention in studies on water, soil and air pollution as a result of the United States Environmental Protection Agency blacklisting 16 PAHs as “priority-controlled pollutants” [4,5]. Currently, many different techniques, such as liquid-phase adsorption, photocatalytic degradation, bioremediation, and electrochemical remediation, have been extensively investigated in treating PAH-contaminated water environments in wastewater reclamation [6,7,8,9]. Among them, adsorption technology seems to be a potential method for PAH control due to its selectivity, low operating cost, affordability, simplicity, high efficiency, and the adsorbent reusability [10,11]. Kumar et al. used pyrolysis-assisted potassium hydroxide-induced palm shell activated carbon to remove PAHs from aqueous solution and the maximum adsorption capacity was 131.7 mg/g [12]. Bhadra et al. proved the adsorption capacity of MOF-derived carbons on naphthalene (237 mg/g), anthracene (284 mg/g), and pyrene (307 mg/g) [13].

Among all kinds of adsorbents, graphene has a high specific surface area, strong hydrophobicity and a unique delocalized large π bond, lending it broad application prospects in the adsorption and treatment of aromatic pollutants from wastewater [14]. Huang et al. [15] synthesized a reduced graphene oxide-hybridized polymeric high-internal phase emulsion with an open-cell structure and hydrophobicity to absorb PAH (47.5 mg/g). However, the complex preparation process, limited adsorption capacity and high cost remain significant obstacles to the large-scale application of graphene in wastewater treatment [16]. Sun et al. compared the adsorption capacity of graphene oxide (GO), reduced graphene oxide (rGO) and graphite (G) for naphthalene, anthracene and pyrene in aqueous solution, and the rGO had the optimal adsorption capacity [17]. Herein, the development of the excellent performance adsorbents had always been the research hotspot in removing PAHs from wastewater.

The previous research show that the metal cations could be the adsorption-active component which could interact with the aromatic ring [18]. Furthermore, the hybridization of metal cations to the adsorption substrate could form the π- complexation interaction which could efficiently realize the extraction of PAHs [19]. The hybridization of metal cations to the graphene materials could be an innovative technology to improve the strength of π complexation in carbon-based materials [20]. MnO_2_ nanoflowers (MnO_2_ NFs) are regarded as optimal nanostructures because of their rational open structure, increasing the adsorption point between the adsorbent and the contaminant [21,22]. At the same time, due to the stable properties and large specific surface area, graphene oxide (GO) can also serve as a porous backbone to support functional materials, thus leading to much mass loading of MnO_2_ NFs for pollutant removal [23,24]. In addition, there is little literature on the performance and mechanism of PAHs using MnO_2_ NFs/GO synthetic materials.

In this study, Benz[b]fluorathene (BbFA) was chosen as the model PAH. It is one of the carcinogenic PAHs, which was included in the list of carcinogenic 2B carcinogens by the International Agency for Research on Cancer (IARC, World Health Organization) in 2017 [25]. In this work, an innovative preparation method of MnO_2_ nanoflower/graphene oxide (MnO_2_ NF/GO) composites was developed to remove BbFA from wastewater. The objectives of the research are (1) to prepare MnO_2_ NF using a direct reduction of KMnO_4_ with poly-(dimethyl diallyl ammonium chloride) (PDDA) and to synthesize MnO_2_ NF/GO in a hydrothermal reactor; (2) to characterize the physicochemical properties of MnO_2_ NF/GO; and (3) to evaluate the BbFA adsorption performance and mechanism of the MnO_2_ NF/GO composite by batch adsorption experiment.

## 2. Results

### 2.1. Structural and Morphology Characterization of MnO_2_ NF/GO Composites

Based on the SEM and TEM observations and analyses of MnO_2_ NF, the nano-agglomerated structure can be observed in Figure 1a,b, which is consistent with the experimental anticipation and similar research results [26]. As displayed in Figure 1c, the morphological characteristics of MnO_2_ NF/GO composites showed an irregular porous structure with a distributed rippled and crumpled morphology, which may increase the surface area of the adsorbent. The HRTEM image (Figure 1d) demonstrated that the lattice distance of 0.341 nm corresponds to the (002) plane of the as-prepared composite [27]. Notably, the STEM and X-ray elemental mappings (Figure 1e) confirm that the MnO_2_ NFs are homogeneously deposited and distributed into the GO.

The XRD pattern of as-synthesized composites was used to analyze the precise crystal structure and the results are demonstrated in Figure 2a. MnO_2_ NF had weak diffraction peaks ascribed to the fact that MnO_2_ NF possess insufficient crystalline property. The peaks at 2*θ* values of 37.28 and 67.94 in MnO_2_ NF were observed, which could be assigned to the (−111) and (114) planes of the MnO_2_ structure (JCPDS card No: 80-1098) [26]. The diffractogram peak at 2*θ* the value of 23.5 in MnO_2_ NF/GO composites is attributed to the amorphous carbon with low graphitization, corresponding to the highly ordered laminar structure with an interlayer distance of 0.34 nm along with the (002) orientation [28]. The diffraction peak at the 2*θ* value of 45 in MnO_2_ NF/GO composites indicates a short-range order in stacked graphene layers. The results were consistent with the previous SEM results.

As displayed in Figure 2b, the adsorbent pores of MnO_2_ NF and MnO_2_ NF/GO composites mainly contain micro-mesoporous structures (diameter < 2 nm) according to the IUPAC classification [29]. The N_2_ adsorption/desorption isotherms for MnO_2_ NF and MnO_2_ NF/GO composites showed a sharp increase at low relative pressure (*P/P_0_*), consistent with the typical curve (type I and IV) with H4 hysteresis loop, which was the microporous structure characteristic [29]. The detailed textural parameters of the as-synthesized composites are shown in Table 1. The *S_BET_* of MnO_2_ NF/GO composites (694.30 m^2^/g) was larger than that of MnO_2_ NF (87.78 m^2^/g), and the volume adsorbed (*V*_tot_) of MnO_2_ NF/GO composite is higher than MnO_2_ NF at a high *P/P_0_* value. These results suggest that MnO_2_ NF/GO composites could facilitate pollutant adsorption due to the nano-agglomerated structure and the large specific surface area and volume adsorbed of MnO_2_ NF/GO composites.

### 2.2. Chemical Characteristics of MnO_2_ NF/GO Composites

The surface chemical characteristics of as-synthesized composites were further analyzed with the typical XPS spectra. As displayed in Figure 3a, C, O and Mn were the major elemental compositions on the surface of MnO_2_ NF/GO composites. Moreover, the Mn _2p_ spectrum for MnO_2_ NF/GO illustrated the successful fabrication of MnO_2_ NF on the GO surface. The high-resolution O 1s spectrum (Figure 3b) shows peaks at 530.6 and 532.2 eV, attributed to Mn-O-Mn and Mn-O-H bonding [30]. The ratio of Mn-O-Mn/Mn-O-H was 4.05 based on the peak area ratios calculation results. These findings suggest that Mn primarily exists in the oxide form (MnO_2_) on the MnO_2_ NF/GO composites, consistent with experimental expectations. As displayed in Figure 3c, the high-resolution wide-range Mn 2p_1/2_ (652.9 eV) and Mn 2p_3/2_ (641.1 eV) peaks using the XPS best peak fitting with Gaussian modes were caused by the overlap of Mn^3+^ and Mn^4+^ ions [31]. Additionally, the separation value (>5.9 eV) between Mn 2p_3/2_ and Mn 2p_1/2_ was consistent with published reports [32]. The presence of carboxyl group and hydroxyl group was conductive for the pollutant adsorption according to the wide-range C 1s spectrum of MnO_2_ NF/GO composites in Figure 3d.

### 2.3. Effect of Contact Time and Adsorption Kinetics

Figure 4a showed the effect of contact time for BbFA adsorption capacity on as-synthesized composites. It could be seen that BbFA was rapidly adsorbed onto adsorbents during the initial 30 min, which can be explained by the rapid diffusion speed of BbFA due to the higher initial BbFA concentration and the initial sufficient adsorption sites of adsorbents. In addition, the large number of aromatic ring structures of BbFA determine the adsorption rate. The BbFA concentration and diffusion speed decrease continuously with continuous contact reaction, while the BbFA adsorption capacity on as-synthesized composite increased. At the same time, the BbFA adsorption capacity on MnO_2_ NF/GO composites was six times higher than that of MnO_2_ NF due to MnO_2_ NF/GO composites’ larger specific surface area and volume adsorbed. To identify the possible rate-controlling steps and reaction mechanisms in the BbFA adsorption process, the pseudo-first-order model and the pseudo-second-order model were used to simulate the experimental data [33].

The pseudo-first-order model, which is based on solid capacity, was defined as follows (Equation (1)):(1)$$ln(q_{e}-q_{t})=lnq_{e}-k_{1}t$$

The pseudo-second-order model, which predicts the behavior of the whole adsorption range, was defined as follows (Equation (2)):(2)$$\frac{t}{q_{t}}=\frac{1}{k_{2}q_{e}^{2}}+\frac{1}{q_{e}}t$$
where *q*_e_ (mg/g) and *q*_t_ (mg/g) are adsorption capacities at equilibrium and time *t*, respectively. *k*_1_ (1/h) are the rate constants of the pseudo-first-order model, and *k*_2_ (g/(mg·min)) are the rate constants of the pseudo-second-order model, respectively.

Figure 4b and c present the plots for the BbFA adsorption of as-synthesized composites by applying the kinetic models in this study, and the slopes and intercepts of these curves were used to determine the fitting parameters. The calculated constants of the kinetics and the corresponding linear regression correlation are shown in Table 2. The high correlation coefficients value (*R*^2^ > 99%) and the excellent agreement between the experimental (*q*_e_) and calculated values (*q_cal_*) indicate that the pseudo-second-order model resulted in a better fit than the pseudo-first-order model. Therefore, the pseudo-second-order model was more suitable for describing the adsorption of BbFA onto MnO_2_ NF/GO composites, and the critical rate-controlling steps were multiple processes, especially the activated or chemisorption processes [34].

### 2.4. Adsorption Isotherm

The adsorption isotherms were generated by changing the initial concentration of BBFA, and the mechanism is that the higher initial concentration of BBFA provides a prominent driving force to control the resistance of BBFA transfer from liquid to solid part in the adsorption system. As displayed in Figure 4d, the adsorption isotherms showed a sharp initial slope due to the fact that the amount of BbFA could not meet as-synthesized composites’ abundance of available adsorption sites in low equilibrium BbFA concentration, resulting in a weakening maximum adsorption capacity. As the equilibrium BbFA concentration increased further, the maximum adsorption capacity increased gradually as its active sites were gradually occupied by BbFA.

The Langmuir isotherm model assumed monolayer coverage of the adsorbate over a homogenous adsorbent surface [35]. The Freundlich equation described the adsorption from low and medium concentrations, when the monolayer was not filled, and the parameter *n* described the heterogeneity of adsorption sites [36]. In this study, the Langmuir and Freundlich isotherms were used to describe the adsorption isotherm in detail (Figure 4a,e,f). The isotherm models were given by Equations (4) and (5):(3)$$Q_{e}=\frac{Q_{0}K_{L}C_{e}}{1+K_{L}C_{e}}$$
(4)$$lnQ_{e}=lnK_{F}+\frac{1}{n}lnC_{e}$$
where *Q_e_* (mg/g) is the maximum adsorption capacity of adsorbents; *C_e_* (mg/L) is the equilibrium BbFA concentration; *Q*_0_ (mg/g) is the initial adsorption capacity; *K_L_* (L/mg) and *K*_F_ ((mg/g)/(L/mg)1/n) are the Langmuir isotherm constant and Freundlich affinity coefficient, respectively; and *n* is the adsorption intensity.

The isotherm lines, isotherm constants, and correlation coefficients of isotherm models are summarized in Figure 4b,c and Table 3. The Langmuir isotherm model exhibited a better fit to the BbFA adsorption process of the MnO_2_ NF (Figure 5), which indicated that the BbFA adsorption tended to be homogeneous and showed monolayer coverage due to the strong interactions between the surface of MnO_2_ NF and BbFA. Further, the Freundlich model was the best for describing the BbFA adsorption process onto the MnO_2_ NF/GO composites, explaining the complex chemical and multi-layer adsorption process due to the metal oxides’ hybridization in MnO_2_ NF/GO composites. In addition, the Freundlich constant *1/n* values were in the range of 0–1, suggesting that the MnO_2_ NF/GO composites can actively adsorb BbFA. As displayed in Table 3, the maximum adsorption capacities (*Q*_e_) of MnO_2_ NF/GO composites (74.07 mg/g) were higher than those of MnO_2_ NF (9.9 mg/g), which were consistent with 2.3 results.

### 2.5. BbFA Adsorption Mechanism

Despite the destruction of the graphene conjugated structure during the GO oxidation process, GO still retains a unique delocalized π bond and surface hydrophobic properties. The π bond on the MnO_2_ NF/GO surface could form π–π interactions with the aromatic ring of BbFA. Many studies have also shown that π–π interaction was an essential way for adsorbents to adsorb PAHs. GO could be used as a metal oxide carrier to synthesize compounds, the compounds can prevent agglomeration from taking place and form π complexing bonds due to metal oxide doping; the adsorption capacity of BbFA on MnO_2_ NF/GO was improved by π complexation. The hydrophobic properties and multilayer structure characteristics of MnO_2_ NF/GO provide sufficient adsorption sites for BbFA. Many oxygen-containing functional groups (carboxyl and hydroxyl) are introduced into MnO_2_ NF/G during GO synthesis and metal oxide doping. At the same time, the adsorption of BbFA molecules would lead to changes in the morphology of the MnO_2_ NF/G, thus generating new adsorption active sites for BbFA removal. The BbFA adsorption capacity of these active sites still needs to be further studied. In general, the results of batch adsorption experiments and model fitting showed that the adsorption of BbFA onto MnO_2_ NF/G was dominated by chemisorption, and the π-π interaction, π complexation, and hydrophobicity of nanoflowers have played a role in the adsorption of BbFA.

## 3. Methods

### 3.1. Materials

Poly dimethyl diallyl ammonium chloride (PDDA, 20%) and potassium permanganate (KMnO_4_, analytical grade) were purchased from ALADDIN Co. Ltd. (Shanghai, China). The fabrication method of graphene oxide (GO) is provided in the Appendix A. Benzo[*b*]fluorathene (BbFA) solid (purchased from Aladdin Industrial Corporation) was of 98% purity. Benzo[*b*]fluorathene (BbFA) was analyzed with GC-MS (please refer to Appendix A for details of method parameters). Additionally, all solutions necessary for analytical procedures were prepared with distilled water, and all the chemicals used were of analytical grade.

### 3.2. Fabrication of MnO_2_ NF/GO Composites

As displayed in Figure 6, the 4.5 mL PDDA was mixed with 20 mL ultrapure water and heated to 120 °C. Afterward, 4.0 g of KMnO_4_ was added to the mixed solution while stirring at 220 rpm for 60 min until the aqueous dispersion mixture turned dark brown, which was defined as MnO_2_ nanoflower aqueous dispersion. The partial MnO_2_ nanoflower aqueous dispersion was centrifuged at 8000 rpm for 10 min and further washed twice with ethanol and three times with distilled water, respectively. Furthermore, the resultant product was dried at 60 °C for 12 h to obtain MnO_2_ nanoflower particles, which were defined as MnO_2_ NFs. The rest of the MnO_2_ nanoflower aqueous dispersion was mixed with the prepared GO (2 g) under continuous stirring at room temperature for 10 min. Then, the resultant mixture was transferred into a hydrothermal reactor and heated to 120 °C for 4 h. Subsequently, the dark precipitate powders were collected and washed with distilled water several times. Finally, they were dried at 60 °C for 12 h to obtain the MnO_2_ NF/GO composite.

### 3.3. Characterization Methods

The characterization methods are provided in the Appendix A.

### 3.4. Adsorption Experiments

The kinetic experiments were conducted to investigate the effect of contact time and evaluate the kinetic properties. The as-synthesized composites (0.2 g) were added into 1 L of BbFA solution with initial concentrations of 0.3 mg/L. The mixture solution had natural pH, which was detected using a pH meter (Model PHS-3C, Shanghai, China). The mixture solution was agitated on the magnetic stirrers (Model 78-1) at a 250 ± 10 rpm speed with control of 25 ± 1 °C. The flasks were wrapped in aluminum foil to prevent photolysis. The 5 ± 0.5 mL samples were taken and filtered at desired adsorption duration (0–240 min), then the mixture was filtered from the liquid phase using a Millipore membrane filter (0.45 μm), and the residual BbFA concentrations were enriched into 10 mL CH_2_Cl_2_ through solid-phase extraction, followed by concentration determination of BbFA using GCMS.

The adsorption capacities of adsorbent were calculated using the following (Equation (5)):*Q = (C_0_ − C_e_)V/M*(5)
where *Q* (mg/g) represents the remove capacities; *C_0_* and *C_e_* (mg/L) are the initial and equilibrium concentrations of BbFA, respectively; *V* (L) is the volume of the BbFA solution; and *M* (g) is the mass of adsorbent added.

In the batch adsorption experiments, the stock solution of BbFA (1 g/L) was prepared by dissolving 0.05 g of powder BbFA in a 500 mL CH_2_Cl_2_ solution, and the desired concentrations were obtained by dilution, followed by magnetic stirring to ensure the complete dissolution of BbFA in water solutions. The batch equilibrium BbFA adsorption studies were performed with a series of brown conical flasks (500 mL) containing a volume of 100 mL of the fixed initial concentration of BbFA (0-350 mg/L). Subsequently, the adsorbents (20 mg) were added to each flask, and the flasks were shaken at 200 ± 10 rpm in a shaded water bath shaker (SHZ-88) at 25 ± 1 °C for 24 h until the equilibrium achieved. The residual BbFA concentration was analyzed using the same method described above.

## 4. Conclusions

MnO_2_ NF aqueous dispersion composed of PDDA and KMnO_4_ was used to produce MnO_2_ NF/GO in a hydrothermal reactor. MnO_2_ NF/GO composites showed excellent removal performance of BbFA from wastewater. The batch adsorption experiments revealed that the adsorption isotherms agreed well with Freundlich isotherm and kinetics obeyed the pseudo-second-order kinetics model and adsorption capacity of 74.07 mg/g. The result was related to the well-developed physicochemical properties of MnO_2_ NF/GO composites. The first reason is that it has a larger specific surface area and adsorption sites, and another important reason is that it has strong electron donor–acceptor interaction (EDA interaction, especially π–π interaction and π complexation). Thus, MnO_2_ NF/GO composites could be cost-effective functional materials for BbFA removal. However, further studies are required to improve oxidative degradation of MnO_2_ NF/GO composites.

## Figures and Tables

**Figure 1 materials-14-04402-f001:**
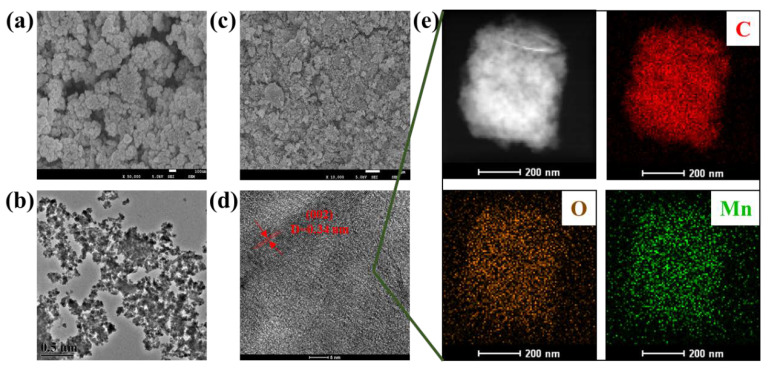
SEM (**a**) and TEM (**b**) images of MnO_2_ NF; SEM (**c**) and HRTEM (**d**) images of MnO_2_ NF/GO composites; and (**e**) HAADF-STEM image of MnO_2_ NF/GO composites and corresponding elemental mapping images of C, O and Mn.

**Figure 2 materials-14-04402-f002:**
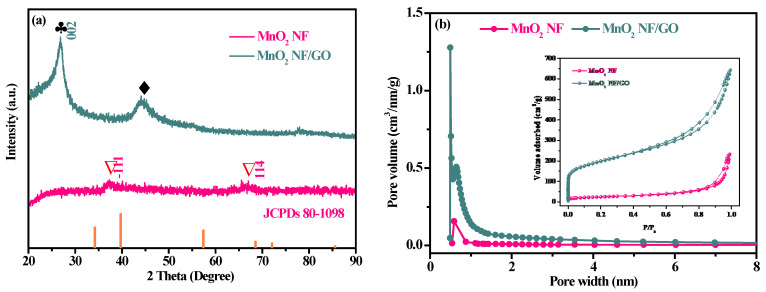
XRD (**a**); the pore size distributions and N_2_ adsorption/desorption isotherms (**b**) of as-synthesized composites.

**Figure 3 materials-14-04402-f003:**
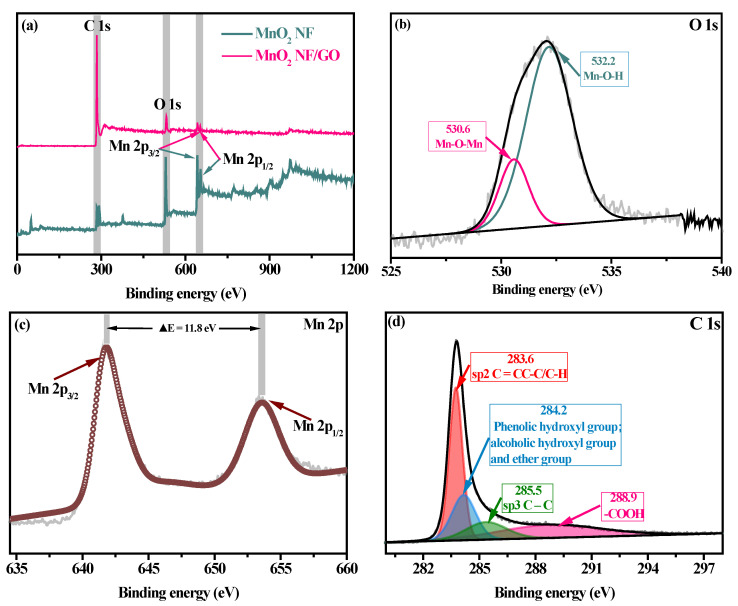
XPS survey spectra of MnO_2_ NF and MnO_2_ NF/GO composites (**a**); O 1s (**b**); Mn 2p (**c**); and C 1s (**d**) spectra of MnO_2_ NF/GO composites.

**Figure 4 materials-14-04402-f004:**
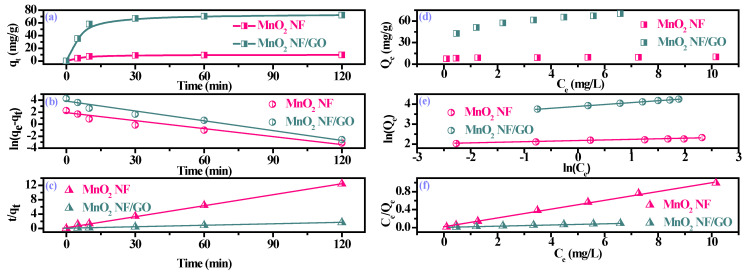
Effect of contact time for BbFA adsorption on as-synthesized composites (**a**); the linear plots of the pseudo-first-order model (**b**) and the pseudo-second-order model (**c**); effect of initial concentration for BbFA adsorption on the as-synthesized composites (**d**); the linear plots of Langmuir isotherm (**e**) and Freundlich isotherm (**f**). (dosage = 0.2 g/L, contact time = 2 h; temperature = 25 ± 1 °C).

**Figure 5 materials-14-04402-f005:**
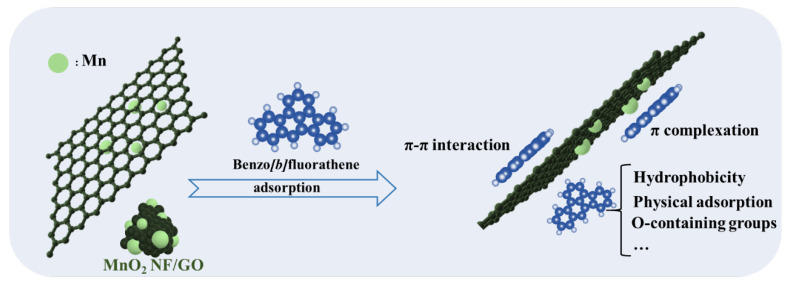
BbFA adsorption mechanism on MnO_2_ NF/GO composites.

**Figure 6 materials-14-04402-f006:**
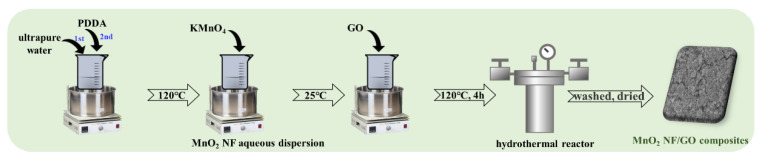
Fabrication of MnO_2_ NF/GO composites.

**Table 1 materials-14-04402-t001:** Textural parameters of as-synthesized composites.

Identification	MnO_2_ NF	MnO_2_ NF/GO
S_BET_ *^[a]^* (m^2^/g)	87.78	694.30
S_mic_ *^[b]^* (m^2^/g)	11.40	182.19
S_ext_ *^[c]^* (m^2^/g)	76.38	512.11
V_tot_ *^[d]^* (cm^3^/g)	0.3584	0.9606
V_mic_ *^[e]^* (cm^3^/g)	0.0046	0.0778
V_mes_ *^[f]^* (cm^3^/g)	0.3538	0.8828

*^[a]^* BET surface area, *^[b]^* micropore surface area, *^[c]^* external surface area, *^[d]^* total pore volume, *^[e]^* micropore volume, *^[f]^* external volume.

**Table 2 materials-14-04402-t002:** Parameters of kinetics models for the BbFA adsorption by as-synthesized composites.

Kinetic Models	Constants	MnO_2_ NF	MnO_2_ NF/GO
Pseudo-first-order parameters	*Q*_e,cal_ (mg/g)	5.272	40.62
	*K*_1_ (1/min)	0.042	0.053
	*R* ^2^	0.9522	0.9721
Pseudo-second-order parameters	*Q*_e,cal_ (mg/g)	9.9	74.07
	*K*_2_ (g/mg/min 10^−4^)	299	48.73
	*R* ^2^	0.998	0.9986

**Table 3 materials-14-04402-t003:** Langmuir and Freundlich constants related to the adsorption isotherms of BbFA for as-synthesized composites.

Isotherm Models	Constants	MnO_2_ NF	MnO_2_ NF/GO
Langmuir	*K_L_* (L/mg)	5.364	74.07
	*Q_m_* (mg/g)	10.13	1.824
	*R* ^2^	0.9973	0.9964
Freundlich	*K_F_* (mg/g·(L/mg)·1/n)	8.701	49.11
	*1/n*	0.0582	0.1878
	*R* ^2^	0.9647	0.9976

## Data Availability

Data sharing is not applicable to this article.

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
