# Peer review of "Removal Performance and Mechanism of Benzo(b)Fluorathene Using MnO2 Nanoflower/Graphene Oxide Composites"

_materials, 2021, doi:10.3390/ma14164402_

Round 1
Reviewer 1 Report
see uploaded doc.

Author Response
对审稿人 1 条评论的回复
评论 1 :
|
是的 |
可以修改 |
必须修改 |
不适用 |
|
引言是否提供了特定的背景并包括所有相关参考资料? |
X |
|
|
|
|
研究设计是否合适? |
X |
|
|
|
|
是否充分描述了这些方法? |
X |
|
|
|
|
结果是否清晰? |
X |
|
|
|
|
结果是否支持结论? |
X |
|
|
|
|
问:谢谢你的评论。
给作者的意见和建议:题为“Removal performance and mechanism of Benzo[b]fluorathene using MnO2 nanoflower/grapheneoxidecomposites”的文章展示了如何以相对简单的方式去除对环境有害的多环芳烃。作者非常清楚地解释了吸收物-吸附剂界面处发生的物理化学现象。
答:谢谢你的评论。
评论 2 :唯一的阅读难点是在引言中宣布的,然后是手稿各个部分的颠倒顺序。
Answer: Thank you for the comment. We have already modified the language of the manuscript with the help of a professional English teacher. And we have already revised order of the individual sections of the manuscript.
Comment 3: A brief summary: Polycyclic aromatic hydrocarbons (PAHs) pose a great threat to human life and health. In the presented article entitled "Removal performance and mechanism of Benzo[b]fluorathene using MnO2 nanoflower/graphene oxid composites", we learn that PAHs removal is possible with the MnO2 NF/GO composite. The paper describes the method of preparation of the MnO2 NF/GO composite and its phytochemical and sorption properties against Benzo[b]fluorathene (BbFA).
答:谢谢你的评论。
评论 4:广泛评论:氧化石墨烯 (GO) 可再生材料,例如用于制造多价金属离子。然而,GO 最常用于制备纳米复合材料。 在此,学习了如何以 相对简单的方式方式获得基于的MnO 2 NF的复合材料。作者使用以下研究技术广泛描述了二氧化锰NF / GO复合材料的结构特性:SEM,TEM,HRTEM,XRD,XPS他们还研究了苯并[b]荧烷( BbFA) 在获得的 MnO 2 NF/GO 复合材料上的动力学、等温线和秘密机制。 提交审稿的文章唯一的论点是手稿各部分的顺序,如引言中所可能的那样,然后在文章中颠倒了。
问:谢谢你的评论。我们已经修改了手稿各部分的发言。
评论5:具体评论: 论文被接受,没有任何进一步的修改。
问:谢谢你的评论。

Reviewer 2 Report
The presented article entitled "Removal performance and mechanism of Benzo[b]fluorathene using MnO2 nanoflower /graphene oxide composites" shows how to remove polycyclic aromatic hydrocarbons that are harmful to the environment in a relatively simple way. The authors explain very clearly the physicochemical phenomena occurring at the absorbate-adsorbent interface.
The only difficulty in reading was announced in the introduction, followed by the reversed order of the individual sections of the manuscript.

Author Response
对审稿人 2 评论的回复
评论 1:需要适度的英语更改。
答:谢谢你的评论。我们已经在专业英语老师的帮助下修改了稿件的语言。
评论2:
|
是的 |
可以改进 |
必须改进 |
不适用 |
|
引言是否提供了足够的背景并包括所有相关参考资料? |
|
X |
|
|
|
研究设计是否合适? |
|
X |
|
|
|
是否充分描述了这些方法? |
|
X |
|
|
|
结果是否清晰呈现? |
|
X |
|
|
|
Are the conclusions supported by the results? |
|
X |
|
|
|
Answer: Thank you for the comment. We have supplied and rewritten the references, methods, and results according to your comment. All the amendments are highlighted in red in the revised manuscript.
Comments and Suggestions for Authors
The manuscript is a valuable contribution to the area, a topic which occupies and will occupy generations of researchers [1]. It is internationally agreed that nanotechnology is a big revolution in production. Nanomaterials at present are being used in as electronics and cosmetics. Many other sectors for example polymeric composite materials are carrying out a lot of scientific and technological works and there are also plans for the wide range of projects being the nanomaterials in use. It has caught tremendous attention and interest for the promotion of nanostructured coatings. All this is because of the unique properties that are at hand, offering possibilities of multifunctionality, reduction of thickness and great spectrum of applications relating technology. However, recent works on nanoparticles introduced in thin layer showcase potential risks of nanoparticle aerosols releases and allow a more balanced benefit/risk analysis. For example, many studies highlight nanoparticle emissions due to coatings, paints [2], tiles [3]. Cases of nanoparticle exposure in the field of occupational hygiene at coating workplaces have been reported [4]. This situation leads to new metrological approach to have a better nanoparticle characterization using TEM [5-7]. Regarding your extensive use of TEM in your manuscript, this approach could be clearly mentioned in your paper.
Comment 3: The overall impression of the manuscript is good. It is well-structured and easily readable. However, a last grammar and vocabulary check by a native English speaker would be appropriate.
Answer: Thank you for the comment. We have already modified the language of the manuscript with the help of a professional English teacher.
Comment 4: A general weakness of the study might be the few number of figures to illustrate your interesting article. You should try to add one Pie chart in your article for example in one chapter of your choice which could help the reader to understand deeply your text.
Answer: Thank you for the comment. All the amendments are highlighted in red in the revised manuscript.
Comment 5: Secondly, references are missing. You should cut-and-paste related references below and put then in your text to improve the article. Moreover, some references from this journal can be added.
Answer: Thank you for the comment. We have supplied and rewrite the references according to your comment.
Comment 6: Title: The title is well chosen and indicates clearly the scope and topic of your contribution.
Answer: Thank you for the comment.
Comment 7: Graphical abstract: “A Graphical Abstract is a single, concise, pictorial and visual summary of the main findings of the article. This could either be the concluding figure from the article or a figure that is specially designed for the purpose, which captures the content of the article for readers at a single glance. Please see examples below.” I this perspective, your graphical abstract could be improved and become more ‘catchy’ and self-explaining.
Answer: Thank you for the comment. We have supplied the Graphical abstract (Figure 1).
Figure 1 Graphical abstract of the manuscript.
Comment 8: Abstract: ‘An abstract is a brief summary of a research article, thesis, review, conference proceeding, or any in-depth analysis of a particular subject and is often used to help the reader quickly ascertain the paper's purpose.’ In this view, your abstract could be improved, similarly to the previous point: 1) By more clearly indicating the purpose of your research. The need for research in this field is certainly a strength of your paper, so tell people about; 2) You might give a hint to the quality of your results, i.e. relative improvement, repeatability etc. The abstract remains vague with respect to this.
Answer: Thank you for the comment. All the amendments are highlighted in red in the revised manuscript.
Comment 9: Introduction: ‘The introduction leads the reader from a general research issue or problem to your specific area of research. It puts your research question in context by explaining the significance of the research being conducted. This is usually done by summarizing current understanding (research to date) and background information about the topic. This is followed by a statement of the purpose of your research issue or problem. This is sometimes followed by a hypothesis or a set of questions you attempt to answer in your research. You may also explain your methodology (how you will research this issue) and explain what your study can reveal. It also may contain a summary of the structure of the rest of the paper.’ The introduction as a whole is corresponding to these needs and very well written. With respect to what has been said before, a minor point would be to introduce the different release scenarios in the framework of risk analysis.
Answer: Thank you for the comment. All the amendments are highlighted in red in the revised manuscript.
Comment 10: Results: Your description of your results is clear and understandable. However, some graphics are poorly defined (see fig 2) and some figures need seeable scale (see a) & c) of fig 1.
Answer: Thank you for the comment. All the amendments are highlighted in red in the revised manuscript.
Comment 11: Conclusions: The conclusions chapter reminds well the reader of the strengths of your central points and summarizes the evidence supporting these.
Answer: Thank you for the comment.
Comment 12:
Suggested references
[1] D.B. Warheit, Hazard and risk assessment strategies for nanoparticle exposures: how far have we come in the past 10 years?, F1000Res, 7 (2018) 376.
[2] M. Morgeneyer, O. Aguerre-Chariol, C. Bressot, STEM imaging to characterize nanoparticle emissions and help to design nanosafer paints, Chemical Engineering Research and Design, 136 (2018) 663-674.
[3] C. Bressot, A. Aubry, C. Pagnoux, O. Aguerre-Chariol, M. Morgeneyer, Assessment of functional nanomaterials in medical applications: can time mend public and occupational health risks related to the products’ fate?, Journal of Toxicology and Environmental Health, Part A, (2018) 1-17.
[4] C. Bressot, N. Shandilya, T. Jayabalan, G. Fayet, M. Voetz, L. Meunier, O. Le Bihan, O. AguerreChariol, M. Morgeneyer, Exposure assessment of Nanomaterials at production sites by a Short Time Sampling (STS) approach Strategy and first results of measurement campaigns, Process Safety and Environmental Protection, (2018).
[5] M. Xiang, O. Aguerre-Chariol, M. Morgeneyer, F. Philippe, Y. Liu, C. Bressot, Uncertainty assessment for the airborne nanoparticle collection efficiency of a TEM grid-equipped sampling system by Monte-Carlo calculation, Advanced Powder Technology, (2021).
[6] M. Xiang, M. Morgeneyer, F. Philippe, M. Manokaran, C. Bressot, Comparative review of efficiency analysis for airborne solid submicrometer particle sampling by nuclepore filters, Chemical Engineering Research and Design, 164 (2020) 338-351.
[7] M. Xiang, M. Morgeneyer, O. Aguerre-Chariol, F. Philippe, C. Bressot, Airborne nanoparticle collection efficiency of a TEM grid-equipped sampling system, Aerosol Science and Technology, (2021) 1-19.
Answer: Thank you for the comment. We have supplied and rewrite the references, methods, and results according to your comment in Supporting Information. The morphology of as-prepared MnO2 NF and MnO2 NF/GO composite were observed by field emission scanning electron microscopy (JEOL 7800 FESEM/SEM) and microstructures were identified by high-resolution transmission electron microscopy (HRTEM, Zeiss Libra 200). 2-4
2 M.Xiang、O.Aguerre-Chariol、M. Morgeneyer、F.Philippe、Y. Liu、C. Bressot,通过蒙特卡罗计算对配备 TEM 网格的采样系统的空中纳米颗粒收集效率的不确定性评估,高级粉末技术,(2021 年)。
3 M.Xiang、M. Morgeneyer、F. Philippe、M. Manokaran、C. Bressot,核孔过滤器对空气中固体亚微米颗粒采样效率分析的比较审查,化学工程研究与设计,164 (2020) 338-351。
4 M.Xiang、M. Morgeneyer、O.Aguerre-Chariol、F.Philippe、C. Bressot,配备 TEM 网格的采样系统的空中纳米颗粒收集效率,气溶胶科学与技术,(2021) 1-19。

Reviewer 3 Report
- «Among them, adsorption technology seems like a potential method for PHAs control due to its selectivity, low operating cost, affordability, simplicity, high efficiency…». References must be provided for each statement.
- «…and the adsorbent reusability can be conducted at room temperature and atmospheric pressure…». What is meant by the term "reusability"? Desorption, regeneration or something else? Desorption, regeneration and subsequent reuse of sorbents in most cases is carried out under the same conditions.
- Page 2, second paragraph from the bottom. The text is strange, contains unstructured information, as well as strange terms.
- «…π complexation refers to forming an π complexation bond between the adsorbate and metal or metal cation and the adsorption process…» The term "Pi-complexation" is a well-known fact and has nothing to do with the adsorption process, there is no need to provide a reference. Pi interactions are a type of non-covalent interaction involving pi systems. In this case, it is assumed that a cation or a neutral metal atom (manganese) will interact with the aromatic ring, which is the pi-system. There is no talk of any adsorption. Please rewrite the paragraph to reflect this comment.
- «…The previous results showed that the hybrid of metal cation…» A hybrid of what?
- «… metal cation could significantly improve the adsorption capacity of PAHs…» Similar to the previous remark, the Authors should rewrite the introduction to indicate that the cation is a sorption-active component. The deposite of a sorption-active component to the matrix leads to the sorbent production for the extraction of organic pollutants due to the pi-interaction.
- «…heterozygous metal ions…» what is this term?
- «Hierarchical architecture MnO2 nanoflowers (MnO2 NFs) have been regarded as optimal nanostructures because of the rational open structure, increasing the adsorption point between the adsorbent and the contaminant..» What do "hierarchical architecture" and "nanoflowers" mean? In addition, what modification of MnO2 are we talking about, todorokite, birnessite, cryptomelane, nsutite, etc. For example, birenessite represents [MnO6] layers between which exchange cations are located.
- «However, the fabrication of MnO2 NFs and GO still faces a challenge owing to the fast and complicated growing MnO2 NFs». It is required to clarify in the text what kind of “growing MnO2 NFs” is meant.
- Page 2, first paragraph from the bottom. The text describing the hazards of benz [b] fluorathene should be moved to the beginning of the section.
- «…flower-like morphology with the nano-agglomerated structure were observed in Figure 1a and .». The authors cite the term "flower-like morphology", but I don't quite understand what criteria were used for the assessment? In my opinion, in Figure 1a and 1b, the sample surface is formed only by particles agglomerations; there is no “flower-like morphology”.
- «As displayed in Figure 1c, the morphological characteristics of MnO 2 NF/GO composites showed an irregular porous structure with a distributed rippled and crumpled morphology, which may increase the surface area of the adsorbent». Figure 1c shows an image of the sample surface. It is incorrect to estimate the porosity of a sample from the surface image; for this, there is a method of low-temperature adsorption of nitrogen.
- Figure 4. Figures are small and not informative, the initial data are not shown, which complicates the understanding of the results. Authors should use the original kinetic equations and adsorption isotherms (in a non-linear type) to describing the initial data using non-linear regression. This will reduce the number of drawings and increase the information content.
- «By contrast, Freundlich isotherm explains the reversible multi-layer adsorption on adsorbent surfaces».
Probably, the authors do not understand the physical meaning of Freundlich model equation, which is a special case of Langmuir model. The Langmuir equation describes the monolayer adsorption process, but not «multi-layer adsorption». If Langmuir model described, while filling by monolayer, the isotherm reached to the maximum and become parallel to X-axis, and Freundlich model described only the initial part of this curve. It is known, that the Henry equation described the adsorption in the ultra-low concentration range: ADS=Kg*C, where Kg – Henry constant, showed the curve slope, and C- it is an equilibrium concentration. If Henry equation will change a little bit, we will get Freundlich equation ADS=Kf*C^n, where Kf –it is Freundlich constant, described the same incline, but the curve’s incline, and degree n – described the deviation from the straight line, those from Henry equation. The Freundlich equation described the adsorption from low and medium concentrations, when the monolayer was not filled, and the parameter n described the heterogeneity of adsorption sites. With multilayer adsorption, the isotherm will be S-curve (S-type), which, for example, can be described with BET equation. For reference, I am citing the article as an example. (1. Saadi, R.; Saadi, Z.; Fazaeli, R.; Fard, N.E. Monolayer and Multilayer Adsorption Isotherm Models for Sorption from Aqueous Media. Korean J. Chem. Eng. 2015, 32, 787–799, doi:10.1007/s11814-015-0053-7.).
- «2.6. Adsorption properties of several adsorbents».
The indication that MnO2 NF / GO composites are the most efficient (Table 4) is not entirely correct. The results of other authors shown in Table 4 were obtained under various conditions such as solution temperature, solid / liquid ratio, stirring speed and duration, etc. Section 2.6 and Table 4 should be deleted.
Author Response
Responses to Reviewer 3’s comments
Comment 1:
|
Yes |
Can be improved |
Must be improved |
Not applicable |
|
Does the introduction provide sufficient background and include all relevant references? |
|
|
X |
|
|
Is the research design appropriate? |
|
X |
|
|
|
Are the methods adequately described? |
|
|
X |
|
|
Are the results clearly presented? |
|
X |
|
|
|
Are the conclusions supported by the results? |
|
X |
|
|
|
Answer: Thank you for the comment. We have supplied and rewrite the references, methods, and results according to your comment. All the amendments are highlighted in red in the revised manuscript.
Comments and Suggestions for Authors
Comment 2: “Among them, adsorption technology seems like a potential method for PHAs control due to its selectivity, low operating cost, affordability, simplicity, high efficiency…”. References must be provided for each statement.
Answer: Thank you for the comment. We have rewritten the expression the paragraph according to your comment.
Among them, adsorption technology seems like a potential method for PHAs control due to its selectivity, low operating cost, affordability, simplicity, high efficiency, and the adsorbent reusability. 10-11
- Yin, W.; Guo, Z.; Zhao, C.; Xu, J., Removal of Cr(VI) from aqueous media by biochar derived from mixture biomass precursors of Acorus calamus Linn. and feather waste. Journal of Analytical & Applied Pyrolysis 2019.
- Yin, W.; Zhao, C.; Xu, J., Enhanced adsorption of Cd (II) from aqueous solution by a shrimp bran modified Typha orientalis biochar. 2019.
Comment 3: “…and the adsorbent reusability can be conducted at room temperature and atmospheric pressure…”. What is meant by the term "reusability"? Desorption, regeneration or something else? Desorption, regeneration and subsequent reuse of sorbents in most cases is carried out under the same conditions.
Answer: Thank you for the comment. We have rewritten the expression the paragraph according to your comment.
Among them, adsorption technology seems like a potential method for PHAs control due to its selectivity, low operating cost, affordability, simplicity, high efficiency, and the adsorbent reusability.
Comment 4: Page 2, second paragraph from the bottom. The text is strange, contains unstructured information, as well as strange terms. “…π complexation refers to forming an π complexation bond between the adsorbate and metal or metal cation and the adsorption process…” The term "Pi-complexation" is a well-known fact and has nothing to do with the adsorption process, there is no need to provide a reference. Pi interactions are a type of non-covalent interaction involving pi systems. In this case, it is assumed that a cation or a neutral metal atom (manganese) will interact with the aromatic ring, which is the pi-system. There is no talk of any adsorption. Please rewrite the paragraph to reflect this comment.
Answer: Thank you for the comment. We have rewritten the expression the paragraph according to your comment.
The previous results showed that the metal cation could be the adsorption-active component which could interact with the aromatic ring. 18 And the hybrid of metal cation to the adsorption substrate could form the π- complexation interaction which could efficiently realize the extraction of PAHs. 19 The hybrid of metal cation to the graphene materials could be innovative technologies to improve the strength of π complexation in carbon-based materials. 20
- Zhang, W.; Zheng, J.; Zheng, P.; Tsang, D.; Qiu, R., The roles of humic substances in the interactions of phenanthrene and heavy metals on the bentonite surface. Journal of Soils and Sediments 2015, 15 (7), 1463-1472.
- Liang, X.; Zhu, L.; Zhuang, S., Sorption of polycyclic aromatic hydrocarbons to soils enhanced by heavy metals: perspective of molecular interactions. Journal of soil & sediments 2016, 16 (5), 1509-1518.
- Takahashi, A.; Yang, R. T., New adsorbents for purification: Selective removal of aromatics. Aiche Journal 2002, 48 (7), 1457-1468.
Comment 5: “…The previous results showed that the hybrid of metal cation…” A hybrid of what? “… metal cation could significantly improve the adsorption capacity of PAHs…” Similar to the previous remark, the Authors should rewrite the introduction to indicate that the cation is a sorption-active component. The deposite of a sorption-active component to the matrix leads to the sorbent production for the extraction of organic pollutants due to the pi-interaction.
Answer: Thank you for the comment. We have rewritten the expression the introduction according to your comment.
The previous results showed that the metal cation could be the adsorption-active component which could interact with the aromatic ring. 18 And the hybrid of metal cation to the adsorption substrate could form the π- complexation interaction which could efficiently realize the extraction of PAHs. 19 The hybrid of metal cation to the graphene materials could be innovative technologies to improve the strength of π complexation in carbon-based materials. 20
- Zhang, W.; Zheng, J.; Zheng, P.; Tsang, D.; Qiu, R., The roles of humic substances in the interactions of phenanthrene and heavy metals on the bentonite surface. Journal of Soils and Sediments 2015, 15 (7), 1463-1472.
- Liang, X.; Zhu, L.; Zhuang, S., Sorption of polycyclic aromatic hydrocarbons to soils enhanced by heavy metals: perspective of molecular interactions. Journal of soil & sediments 2016, 16 (5), 1509-1518.
- Takahashi, A.; Yang, R. T., New adsorbents for purification: Selective removal of aromatics. Aiche Journal 2002, 48 (7), 1457-1468.
Comment 6: “…heterozygous metal ions…” what is this term? "
Answer: Thank you for the comment. We have rewritten the expression the term according to your comment.
The previous results showed that the metal cation could be the adsorption-active component which could interact with the aromatic ring. 18 And the hybrid of metal cation to the adsorption substrate could form the π- complexation interaction which could efficiently realize the extraction of PAHs. 19 The hybrid of metal cation to the graphene materials could be innovative technologies to improve the strength of π complexation in carbon-based materials. 20
- Zhang, W.; Zheng, J.; Zheng, P.; Tsang, D.; Qiu, R., The roles of humic substances in the interactions of phenanthrene and heavy metals on the bentonite surface. Journal of Soils and Sediments 2015, 15 (7), 1463-1472.
- Liang, X.; Zhu, L.; Zhuang, S., Sorption of polycyclic aromatic hydrocarbons to soils enhanced by heavy metals: perspective of molecular interactions. Journal of soil & sediments 2016, 16 (5), 1509-1518.
- Takahashi, A.; Yang, R. T., New adsorbents for purification: Selective removal of aromatics. Aiche Journal 2002, 48 (7), 1457-1468.
Comment 7: “Hierarchical architecture MnO2 nanoflowers (MnO2 NFs) have been regarded as optimal nanostructures because of the rational open structure, increasing the adsorption point between the adsorbent and the contaminant.” What do "hierarchical architecture" and "nanoflowers" mean? In addition, what modification of MnO2 are we talking about, todorokite, birnessite, cryptomelane, nsutite, etc. For example, birenessite represents [MnO6] layers between which exchange cations are located.
Answer: Thank you for the comment. We have rewritten the inappropriate expression according to your comment. And MnO2 nanoflowers (MnO2 NFs) is a description of the morphology and structure of manganese dioxide. In addition, this study focused on MnO2-modified GO, so todorokite, birnessite, cryptomelane and nsutite were not discussed.
MnO2 nanoflowers (MnO2 NFs) have been regarded as optimal nanostructures because of the rational open structure, increasing the adsorption point between the adsorbent and the contaminant 21-22. At the same time, due to the stable properties and large specific surface area, graphene oxide (GO) can also serve as a porous backbone to support functional materials, thus leading to a much mass loading of MnO2 NFs for pollutant removal 23-24. However, the fabrication of MnO2 NFs and GO still faces a challenge owing to the fast and complicated growing MnO2 NFs. In addition, there is little literature on the performance and mechanism of PHAs using MnO2 NFs/GO synthetic materials.
Fabrication method of MnO2 NF composites:
As displayed in Figure 6, the 4.5 mL PDDA was mixed with 20 mL ultrapure water and heated to 120 °C. Afterward, 4.0 g of KMnO4 was added to the mixed solution while stirring at 220 rpm for 60 min until the mixture aqueous dispersion turned dark brown, were defined as MnO2 nanoflower aqueous dispersion. The partial MnO2 nanoflower aqueous dispersion was centrifuged at 8000 rpm for 10 min and further washed twice with ethanol and three times with distilled water, respectively. Furthermore, the resultant was dried at 60 ◦C for 12 h to get MnO2 nanoflower particles, which were defined as MnO2 NFs.
Figure 6. Fabrication method of MnO2 NF composites.
Comment 8: “However, the fabrication of MnO2 NFs and GO still faces a challenge owing to the fast and complicated growing MnO2 NFs”. It is required to clarify in the text what kind of “growing MnO2 NFs” is meant.
Answer: Thank you for the comment. We have removed the inappropriate description.
Comment 9: Page 2, first paragraph from the bottom. The text describing the hazards of Benz[b]fluorathene should be moved to the beginning of the section.
Answer: Thank you for the comment. We have rewritten the expression the term according to your comment.
In this study, Benz[b]fluorathene was chosen as the model PHA. Benz[b]fluorathene (BbFA) is one of the carcinogenic polycyclic aromatic hydrocarbons, which was included in the list of carcinogenic 2B carcinogens by the International Agency for Research on Cancer (IARC, World Health Organization) in 2017.
Comment 10: “…flower-like morphology with the nano-agglomerated structure were observed in Figure 1a and ”. The authors cite the term "flower-like morphology", but I don't quite understand what criteria were used for the assessment? In my opinion, in Figure 1a and 1b, the sample surface is formed only by particles agglomerations; there is no “flower-like morphology”.
Answer: Thank you for the comment. We have rewritten the expression the term according to your comment.
Based on the SEM and TEM observations and analyses of MnO2 NF, the nano-agglomerated structure was observed in Figure 1a and 1b, which were consistent with the experimental anticipation and similar research results.
Comment 11: “As displayed in Figure 1c, the morphological characteristics of MnO2 NF/GO composites showed an irregular porous structure with a distributed rippled and crumpled morphology, which may increase the surface area of the adsorbent”. Figure 1c shows an image of the sample surface. It is incorrect to estimate the porosity of a sample from the surface image; for this, there is a method of low-temperature adsorption of nitrogen.
Answer: Thank you for the comment.
The purpose of the evaluation of the sample porosity according to the surface image in this paper was first to assume and speculate that the sample maybe have a more developed surface structure, and then to support the sample porosity characteristics of the sample through N2 adsorption/desorption isotherms (in Supporting Information).
Comment 12: Figure 4. Figures are small and not informative, the initial data are not shown, which complicates the understanding of the results. Authors should use the original kinetic equations and adsorption isotherms (in a non-linear type) to describing the initial data using non-linear regression. This will reduce the number of drawings and increase the information content.
Answer: Thank you for the comment. We have revised the Figure 4 according to your comment.
Figure 4. Effect of contact time for BbFA adsorption on as-synthesized composites (a); the linear plots of the pseudo-first-order model (b) and the pseudo-second-order model (c); Effect of initial concentration for BbFA adsorption on the as-synthesized composites (d); the linear plots of Langmuir isotherm (e) and Freundlich isotherm (f). (dosage = 0.2 g/L, contact time = 2h; temperature = 25 ± 1 ℃).
Comment 13: “By contrast, Freundlich isotherm explains the reversible multi-layer adsorption on adsorbent surfaces”. Probably, the authors do not understand the physical meaning of Freundlich model equation, which is a special case of Langmuir model. The Langmuir equation describes the monolayer adsorption process, but not «multi-layer adsorption». If Langmuir model described, while filling by monolayer, the isotherm reached to the maximum and become parallel to X-axis, and Freundlich model described only the initial part of this curve. It is known, that the Henry equation described the adsorption in the ultra-low concentration range: ADS=Kg*C, where Kg – Henry constant, showed the curve slope, and C- it is an equilibrium concentration. If Henry equation will change a little bit, we will get Freundlich equation ADS=Kf*C^n, where Kf –it is Freundlich constant, described the same incline, but the curve’s incline, and degree n – described the deviation from the straight line, those from Henry equation. The Freundlich equation described the adsorption from low and medium concentrations, when the monolayer was not filled, and the parameter n described the heterogeneity of adsorption sites. With multilayer adsorption, the isotherm will be S-curve (S-type), which, for example, can be described with BET equation. For reference, I am citing the article as an example. (1. Saadi, R.; Saadi, Z.; Fazaeli, R.; Fard, N.E. Monolayer and Multilayer Adsorption Isotherm Models for Sorption from Aqueous Media. Korean J. Chem. Eng. 2015, 32, 787–799, doi:10.1007/s11814-015-0053-7.).
Answer: Thank you for the comment. We have rewritten the expression the paragraph according to your comment.
The Langmuir isotherm model assumed monolayer coverage of the adsorbate over a homogenous adsorbent surface. 35 The Freundlich equation described the adsorption from low and medium concentrations, when the monolayer was not filled, and the parameter n described the heterogeneity of adsorption sites. 36
- Marczewski, A. W., Analysis of kinetic Langmuir model. Part I: integrated kinetic Langmuir equation (IKL): a new complete analytical solution of the Langmuir rate equation. Langmuir 2010, 26 (19), 15229-15238.
- Saadi, R.; Saadi, Z.; Fazaeli, R.; Fard, N.E. Monolayer and Multilayer Adsorption Isotherm Models for Sorption from Aqueous Media. Korean J. Chem. Eng. 2015, 32, 787-799.
Comment 14: “2.6. Adsorption properties of several adsorbents”. The indication that MnO2 NF / GO composites are the most efficient (Table 4) is not entirely correct. The results of other authors shown in Table 4 were obtained under various conditions such as solution temperature, solid / liquid ratio, stirring speed and duration, etc. Section 2.6 and Table 4 should be deleted.
Answer: Thank you for the comment. We have deleted the Section 2.6 and Table 4 according to your comment.

Round 2
Reviewer 3 Report
Authors corrected all the comments and made the necessary additions.
- However, the article also contains the term “heterozygosity” (page 8), which got into the text, probably by mistake from the biology.
- In the future, I recommending to Authors use the non-linear regression (Figure 3) to describe experimental data.
Author Response
Responses to Reviewer 1’s comments
Comment 1:
|
Yes |
Can be improved |
Must be improved |
Not applicable |
|
Does the introduction provide sufficient background and include all relevant references? |
X |
|
|
|
|
Is the research design appropriate? |
X |
|
|
|
|
Are the methods adequately described? |
X |
|
|
|
|
Are the results clearly presented? |
X |
|
|
|
|
Are the conclusions supported by the results? |
X |
|
|
|
|
Answer: Thank you for the comment.
Comment 2: Authors corrected all the comments and made the necessary additions. However, the article also contains the term “heterozygosity” (page 8), which got into the text, probably by mistake from the biology.
Answer: Thank you for the comment. We have rewritten the expression the paragraph according to your comment.
Many oxygen-containing functional groups (carboxyl and hydroxyl) are introduced into MnO2 NF/G during GO synthesis and metal oxide doping.
Comment 3: In the future, I recommending to Authors use the non-linear regression (Figure 3) to describe experimental data.
Answer: Thank you for the comment. In the future, we will use the non-linear regression to describe experimental data.
